# Lung cancer associated autoantibody responses are detectable years before clinical presentation

**Jared Allen** [1,2]*, **Graham Healey**[2], **Isabel Macdonald**[2]

**1** Division of Cancer and Stem Cells, University of Nottingham, Nottingham, United Kingdom, **2** Freenome Ltd., Nottingham, United Kingdom

* jared.allen@freenome.com

## Abstract

The EarlyCDT-Lung® test detects elevated levels of tumour-associated autoantibodies generated in response to immune recognition of cancerous cells, and these autoantibodies have previously been shown to precede clinical presentation of lung cancer. Using a longitudinal cohort from the United Kingdom Collaborative Trial of Ovarian Cancer Screening (UKCTOCS) study, we have established that elevated autoantibodies can be detected an average of four years in advance of clinical presentation, and in some cases up to 8 years prior to clinical presentation. This is the first study to establish pre-diagnostic elevation of autoantibodies using samples from a longitudinal prospective clinical trial using a clinically validated and commercially available biomarker panel.

## Introduction

Globally, lung cancer has the highest mortality rate of all cancers, and was estimated to be responsible for nearly 1.8 million deaths in 2020 [1]. Lung cancer is generally not detected until symptomatic presentation, at which point it is usually advanced stage, with extremely poor prognosis. For this reason, detecting lung cancer at an early stage can vastly improve 5-year survival, with stage 1 detected lung cancers having a 5-year survival of 62.7%, compared to only 4.3% for those cancers diagnosed at stage 4 [2].

In vitro estimates of imaging detection limits have suggested that a lung cancer does not become detectable by current imaging modalities until it has reached a population of at least 100,000 cells [3], assuming exponential growth, this would suggest that on average a malignant cell population would need to double 16.6 times ($\log_2(100000)$) before becoming detectable by imaging. Radiography studies have determined a mean doubling time in malignant lung cancer of 158 days [4], which would denote that, on average, a lung cancer is present and potentially able to elicit an autoimmune response for 7.2 years before it can be confirmed by imaging. This is supported by studies that have shown evidence of detectable levels of tumour-associated autoantibodies in individuals prior to presenting with cancer, including p53 responses a median of 3.5 years prior to lung cancer imaging detection [5], and the detection of p53 and Her2 in prediagnostic breast cancer samples [6]. Imaging surveillance of subjects after a positive autoantibody test response should allow for detection of a cancer at the very earliest stages, vastly improving prognosis.

**Data availability statement:** All relevant data are within the paper and its Supporting information files.

**Funding:** The author(s) received no specific funding for this work.

**Competing interests:** The authors have declared that no competing interests exist.

The EarlyCDT-Lung test [7,8] is a simple blood test which detects elevated levels of a panel of tumour-associated autoantibodies generated in response to abnormal tumour cells. This test is CE marked and has been validated in both case-control studies and use in clinical practice [9]. Its use as a screening test in a high-risk population has been shown to result in a stage shift in diagnosis favouring early-stage detection [10]. Previous validation studies have focused on diagnostic sensitivity and specificity at time of testing, but the length of time prior to clinical presentation of lung cancer (lead time) at which elevated autoantibodies can be detected using the EarlyCDT-Lung test has previously not been established.

The (UKCTOCS) [11] was a large prospective trial that aimed to quantify the benefits of an ovarian cancer screening programme. In this trial 202,638 postmenopausal women aged 50 and above were recruited through thirteen UK centres between 2001 and 2005. The multimodal screening arm of this study contained 50,640 women for whom annual blood samples were taken, all of whom were followed up for development of cancer. Within this cohort a number of women went on to develop lung cancer; analysis of their longitudinal blood samples will allow assessment of when elevated autoantibodies were first detectable in their blood samples, compared to the time at which their cancer was diagnosed according to cancer registry data.

The primary aim of the study was to assess the diagnostic performance of the clinically validated EarlyCDT-Lung test over the years prior to confirmed cancer diagnosis, in order to estimate how early before routine symptomatic or incidental clinical presentation tumour-associated autoantibodies measured by the test are elevated and detectable.

## Materials and methods

### Patient cohorts

Case and control cohorts were identified from samples collected as part of the UKCTOCS trial of post-menopausal women aged 50–74 years, and aligned with UKCTOCS trial exclusion criteria (no history of bilateral oophorectomy or ovarian malignancy, no increased risk of familial ovarian cancer, and no active non-ovarian malignancy). The case cohort was comprised of 142 subjects who presented with lung cancer during the course of the UKCTOCS study follow up, and who had at least three serial samples and a known date of lung cancer diagnosis. Subjects were matched for age at trial entry (+/− 5 years), smoking history, and trial entry date (+/− 2 years) to a control cohort of 142 subjects who had no evidence of developing lung cancer during the UKCTOCS follow-up period. All cases had between 4 and 8 longitudinal samples (median 7), covering a period of between 3.1 and 8.9 years (median 6.2), while controls had between 3 and 8 longitudinal samples (median 6) covering a period of between 2.9 and 9.2 years (median 6.3). Demographic details of the cohorts are outlined in Table 1. Histological subtypes in the cancer cases were 49% adenocarcinoma, 17% squamous cell carcinoma, 16.4%

**Table 1. Cohort demographics.**

| Variable | N | CASE, N = 142 | CONTROL, N = 142 | p-value |
|---|---|---|---|---|
| Age at Sample Collection[1] | 284 | 64 (59, 69) | 64 (59, 69) | 0.8[3] |
| Smoking status | 258 | | | 0.9[4] |
| Non-Smoker[2] | | 27 (21%) | 28 (22%) | |
| Smoker[2] | | 102 (79%) | 101 (78%) | |
| Unknown | | 13 | 13 | |

1 Median (IQR), 2 n (%).

3 Wilcoxon rank sum test, 4 Pearson's Chi-squared test.

unspecified non-small cell carcinoma, 13% small cell carcinoma, with the remaining 5.6% comprised of carcinoid tumour, large cell carcinoma, and neuroendocrine carcinoma.

All samples were received blinded and assessed on the EarlyCDT-Lung test for autoantibody levels to a panel of seven tumour-associated antigens, these levels were compared to pre-determined commercial cut-off thresholds to return a panel assessment of either "negative" referring to no elevated risk of lung cancer, "moderate positive" relating to an elevated risk of lung cancer, or "high positive" referring to a highly elevated risk of lung cancer.

Samples were unblinded, and positivity assessed by subject and longitudinal timepoint. For case samples, assay positivity was compared to date of clinical presentation to determine how early a detectable autoantibody response was present, prior to detection by methods available at time of collection.

Additionally, time to detection has been assessed by histological subtype for the three most prevalent subtypes in the dataset: adenocarcinoma (69 subjects, 49% of the cohort), squamous cell carcinoma (24 subjects, 17% of the cohort), and small cell carcinoma (18 subjects, 13% of the cohort).

Finally, a brief investigation was undertaken to compare whether sustained positive responses were more likely to be demonstrated in case samples over controls, with a sustained positive being defined as a subject with two or more consecutive positive (either moderate positive or high positive) EarlyCDT-Lung test results, while subjects without consecutive positive results were considered transient positive results.

### Study conduct and ethics

All analysis was performed on retained anonymized samples from the UKCTOCS trial (ISRCTN Number 22488978), for which informed consent for use in future projects was obtained at time of collection. The detailed study is a subsidiary analysis which forms part of a larger collaborative project between Oncimmune Ltd., Abcodia Ltd., and University College London, to be reported by Healey et al. (publication in draft). As part of the collaborative agreement, ethics approval for the project was obtained by Abcodia from the Abcodia/UCL Steering Committee and the UCL Research Ethics Committee prior to supplying samples to Oncimmune. Ethics approval was obtained Q2 2016, and samples were run between the 9th and 19th August 2016, with corresponding demographic and outcomes data being provided on the 21st September 2016. At no point was information made available that would allow the identification of individual participants.

## Results

### Clinical performance

EarlyCDT-Lung test performance for the study cohorts was assessed over all samples, with subjects being assigned the highest level of positivity returned from any of their longitudinal samples, the results of which are summarised in Table 2, and show that these samples returned a sensitivity of 26.1%, and specificity of 88.7%.

**Table 2. 2 × 2 contingency table summarising diagnostic performance (moderate and high positive) over all samples.**

|  | EarlyCDT-Lung Positive | EarlyCDT-Lung Negative | Total |
|---|---|---|---|
| Cases | 37 (26.1%) | 105 (73.9%) | 142 |
| Controls | 16 (11.3%) | 126 (88.7%) | 142 |
| Total | 53 (18.7%) | 231 (81.3%) | 284 |

## Cancer case time to detection – moderate and high positive.

Examining all positive responses (both moderate positive and high positive) as summarized in Table 3, autoantibodies were detected up to 101.0 months (8.4 years) prior to clinical presentation, with positive responses being present a median of 49.9 months (4.2 years) before presentation of lung cancer. The earliest autoantibody responses were observed against p53, which also showed the highest sensitivity in this study, however responses were observed in all autoantibodies other than HuD at time points in excess of 73 months (6.1 years) prior to clinical presentation with current detection methods.

Table 3. Earliest detection and median time to detection by antigen (EarlyCDT-Lung moderate and high positive).

| Antigen | Number of Cases Positive | Earliest Pre-Dx Time to Detection (months) | Median Pre-Dx Time to Detection (months) |
|---|---|---|---|
| p53 | 15 | 101.0 | 53.3 |
| SOX-2 | 2 | 73.6 | 73.2 |
| CAGE | 6 | 78.6 | 24.1 |
| NY-ESO-1 | 6 | 95.5 | 31.0 |
| GBU 4-5 | 3 | 77.5 | 58.1 |
| MAGE-A4 | 6 | 81.2 | 48.1 |
| HuD | 0 | NA | NA |
| Panel | 37 | 101.0 | 49.9 |

## Cancer case time to detection – high positive

EarlyCDT-Lung high-positive responses are summarized in Table 4 and show high-positive results were detected up to 60 months (5 years) prior to clinical presentation, with a median time of 15.6 months (1.3 years) from high-positive autoantibody test to presentation.

Table 4. Earliest detection and median time to detection by antigen (EarlyCDT high positive only).

| Antigen | Number of Cases Positive | Earliest Pre-Dx Time to Detection (months) | Median Pre-Dx Time to Detection (months) |
|---|---|---|---|
| p53 | 4 | 31.2 | 9.9 |
| SOX-2 | 1 | 60.0 | 60.0 |
| CAGE | 4 | 33.4 | 8.8 |
| NY-ESO-1 | 4 | 50.6 | 33.4 |
| GBU 4-5 | 0 | NA | NA |
| MAGE-A4 | 1 | 14.9 | 14.9 |
| HuD | 0 | NA | NA |
| Panel | 14 | 60.0 | 15.6 |

## Cancer case time to detection by subtype – moderate and high positive

To determine whether appreciable differences are present in time to detection for different lung cancer histological subtypes, time to detection has been calculated for the three most prevalent subtypes in the dataset and detailed in the following Tables 5–7.

**Adenocarcinoma.**

**Table 5. Adenocarcinoma earliest detection and median time to detection by antigen (EarlyCDT moderate and high positive).**

| Antigen | Number of Cases Positive | Earliest Pre-Dx Time to Detection (months) | Median Pre-Dx Time to Detection (months) |
| --- | --- | --- | --- |
| p53 | 5 | 86.6 | 50.1 |
| SOX-2 | 1 | 72.9 | 72.9 |
| CAGE | 2 | 40.9 | 39.8 |
| NY-ESO-1 | 3 | 75.2 | 48.6 |
| GBU 4-5 | 0 | NA | NA |
| MAGE-A4 | 4 | 81.2 | 48.2 |
| HuD | 0 | NA | NA |
| Panel | 37 | 86.6 | 47.4 |

**Squamous cell carcinoma.**

**Table 6. Squamous cell carcinoma earliest detection and median time to detection by antigen (EarlyCDT-Lung moderate and high positive).**

| Antigen | Number of Cases Positive | Earliest Pre-Dx Time to Detection (months) | Median Pre-Dx Time to Detection (months) |
| --- | --- | --- | --- |
| p53 | 5 | 81.5 | 75.4 |
| SOX-2 | 1 | 73.6 | 73.6 |
| CAGE | 3 | 9.6 | 4.0 |
| NY-ESO-1 | 1 | 95.5 | 95.5 |
| GBU 4-5 | 2 | 58.1 | 38.0 |
| MAGE-A4 | 2 | 49.9 | 34.3 |
| HuD | 0 | NA | NA |
| Panel | 14 | 95.5 | 51.6 |

**Small cell carcinoma.**

**Table 7. Small cell carcinoma earliest detection and median time to detection by antigen (EarlyCDT-Lung moderate and high positive).**

| Antigen | Number of Cases Positive | Earliest Pre-Dx Time to Detection (months) | Median Pre-Dx Time to Detection (months) |
| --- | --- | --- | --- |
| p53 | 0 | NA | NA |
| SOX-2 | 0 | NA | NA |
| CAGE | 0 | NA | NA |
| NY-ESO-1 | 1 | 13.4 | 13.4 |
| GBU 4-5 | 1 | 77.5 | 77.5 |
| MAGE-A4 | 0 | NA | NA |
| HuD | 0 | NA | NA |
| Panel | 2 | 77.5 | 45.5 |

**Longitudinal assessment - sustained vs transient positive responses.**

Table 8. Comparison of sustained (2 or more consecutive positive test results) and transient (no consecutive positive test results) positive responses in case and control subjects.

| | Sustained EarlyCDT-Lung Positive | Transient EarlyCDT-Lung Positive | EarlyCDT-Lung Negative | Total |
|---|---|---|---|---|
| Cases | 25 (17.6%) | 12 (8.5%) | 105 (73.9%) | 142 |
| Controls | 8 (5.6%) | 8 (5.6%) | 126 (88.7%) | 142 |
| Total | 33 (11.6%) | 20 (7.0%) | 231 (81.3%) | 284 |

# Discussion

Analysis of the cohort of subjects that went on to develop lung cancer within the UKCTOCS study shows that EarlyCDT-Lung was able to identify tumour associated autoantibody responses up to a maximum of 8.4 years before clinical presentation, with detectable elevated autoantibody responses presenting a median of 4.2 years before detection by CT in subjects that went on to develop lung cancer. This is comparable to the work published by Li et al [5] which reported elevated p53 autoantibodies were detectable an average of 3.5 years prior to clinical presentation in a cohort of 49 high risk asbestosis subjects who subsequently developed a lung cancer, and supports the theory that the immune system responds to the presence of tumour-associated antigens early in cancer development, while the cancer is still comprised of a relatively small number of cells, and detection and monitoring of these autoantibodies represents a promising opportunity to identify a malignancy at its earliest stages.

Observation of the change in autoantibody signal over time (as shown in S1 Appendix) reveals that many of the changes from negative to positive between time points are associated with a drastic increase in autoantibody signal. This reflects the natural amplification of an immune response to a malignancy and lends further credence to these being highly specific responses to antigen mutation or overexpression as a result of tumour growth.

An examination of the samples that showed a maintained positive EarlyCDT-Lung test result over consecutive samples, compared to those which showed only a transient positive result (Table 8), demonstrated that a sustained response is more likely in true positive case results, with 67.6% of true positive results showing sustained positivity compared to only 50% of false positive cases showing positive results sustained over 2 or more consecutive samples.

While the presence of autoantibodies long before imaging detection presents an opportunity to treat a cancer at its earliest stages, treatment is still limited to visible tumours, necessitating a period of surveillance in subjects with a positive test prior to tumour presentation. This has been explored in the ECLS trial [10], a phase IV prospective clinical trial which recruited high risk patients aged between 50 and 75, with either a 20 pack-year smoking history, or a history of smoking and an immediate family history of lung cancer, and employed a strategy of two-year follow up of a positive EarlyCDT result through six monthly low dose CT scans. This trial was able to show a 14.3% absolute risk reduction in late-stage presentation, and a 29.2% reduction in lung cancer mortality. This study also calculated a number needed to screen to prevent one late-stage lung cancer diagnosis 2 years after EarlyCDT screen of 325 patients. A health economics analysis [12] based on the results of this trial showed that low-dose CT monitoring at six monthly intervals for two years following a positive test result was cost effective a less than £20,000 per QALY, showing that such surveillance strategies following EarlyCDT screening are health economically viable.

The cohort used in this study, while larger than those examined in previous studies [5,6], was relatively small, owing to the difficulty of collecting substantial numbers of longitudinal

prediagnostic cancer samples, even in high-risk populations, and this in combination with the high heterogeneity of lung cancer, limits the conclusions drawn at the individual autoantibody level. The results are further complicated as the UKCTOCS collection was primarily concerned with identifying ovarian cancer. Therefore this cohort is not the intended screening population for the EarlyCDT-Lung test, being entirely female and with a large proportion of never smokers (~20%). These factors may contribute to the observed sensitivity of 26.1% (95% confidence intervals 18.8%–33.3%) in this cohort being lower than the 37.1% previously observed during clinical use [9]. Additionally, while the EarlyCDT-Lung test has been validated in case-control studies to have a specificity around 90%, evidence of elevated autoantibodies up to 8.4 years prior to clinical presentation as presented by this analysis raises the possibility that at least a portion of those presenting as false positives in these case-control studies represent latent cancers that may subsequently present, or are potentially indicative of a successful immune response to mutated cells that will resolve without malignant presentation. Follow-up of the control cohort in this study revealed four subjects with a subsequent cancer diagnosis, one (bone cancer secondary to unknown primary) around the time of last sample, and the remaining three (breast, oesophageal, and peritoneal) presenting 5–6 years after the end of the trial, with one additional suspected of a thyroid malignancy around two years after last sample collection. Of the four with confirmed malignancies, one (peritoneal) returned a positive result for MAGE-A4 autoantibodies in the EarlyCDT-Lung test. Disregarding these five subjects when assessing clinical performance in this study would return an increased specificity of 89.1%.

Finally, median times to detection observed did show some differences by autoantibody. This was explored further through analysis by histological subtype, as summarized in Tables 5–7. Tumour doubling times have previously been shown to differ based on histology, with adenocarcinoma typically being associated with longer doubling times than squamous cell carcinoma or small cell carcinoma [13]. However, median time to detection in this study was comparable between adenocarcinoma, squamous cell carcinoma, and small cell carcinoma, at 47.4, 51.6, and 45.5 months respectively. Within adenocarcinoma and squamous cell carcinoma subjects, CAGE autoantibody responses were evident closer to diagnosis than other panel autoantibodies, especially in squamous cell carcinoma. This may be indicative that CAGE overexpression or mutation either develop later in disease progression or are possibly related to more aggressive malignancies with shorter doubling times. While further work is needed on larger cohorts to fully understand these relationships, and better characterize the activity of the immune system in response to early malignancy, these results show the potential that assessment of autoantibodies may have in assessing the presence and aggressiveness of a malignancy prior to its clinical presentation.

Although this study was limited by the size and nature of the examined cohort, the ability to detect autoantibodies to tumour-associated proteins at an average of four years prior to clinical presentation shows that these immune responses occur early in tumour development, and their detection through a simple and accessible blood test has the potential to become a vital component in early lung cancer diagnosis and treatment strategies.

## Supporting information

**S1 Appendix. Longitudinal autoantibody profiles in positive cases, showing earliest positive autoantibody levels, and median time of elevated signal prior to detection.**
(PDF)

**S2 Data. Wide format datafile containing data used in described analyses.**
(CSV)

## Acknowledgments

We would like to thank Professor Usha Menon, Abcodia Ltd., and University College London for facilitating access to the samples used in this study. We would also like to thank Laura Peek and the staff at Oncimmune LLC for performing the assays and providing the raw data for analysis.

## Author contributions

**Conceptualization:** Graham Healey.

**Data curation:** Jared Allen.

**Formal analysis:** Jared Allen.

**Investigation:** Jared Allen.

**Methodology:** Jared Allen.

**Project administration:** Graham Healey.

**Supervision:** Graham Healey, Isabel Macdonald.

**Visualization:** Jared Allen.

**Writing – original draft:** Jared Allen.

**Writing – review & editing:** Graham Healey, Isabel Macdonald.

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
