## [Decision Letter · Decision Letter 0]

15 Jul 2024

PONE-D-24-11623Lung Cancer Associated Autoantibody Responses are Detectable Years Before Clinical PresentationPLOS ONE

Dear Dr. Allen,

Thank you for submitting your manuscript to PLOS ONE. After careful consideration, we feel that it has merit but does not fully meet PLOS ONE’s publication criteria as it currently stands. Therefore, we invite you to submit a revised version of the manuscript that addresses the points raised during the review process. The manuscript lacks novelty but has the capability to build on the foundation knowledge on early CDT. The reviewers 1 and 2 have identified various limitations of the study that need proper justification from the authors.  Kindly review each and every comment of reviewers 1 and 2 and address them accordingly. 

Please submit your revised manuscript by Aug 29 2024 11:59PM. If you will need more time than this to complete your revisions, please reply to this message or contact the journal office at plosone@plos.org . Please include the following items when submitting your revised manuscript:

We look forward to receiving your revised manuscript.

Kind regards,

Afsheen Raza, PhD

Academic Editor

PLOS ONE

2. In this instance it seems there may be acceptable restrictions in place that prevent the public sharing of your minimal data. However, in line with our goal of ensuring long-term data availability to all interested researchers, PLOS’ Data Policy states that authors cannot be the sole named individuals responsible for ensuring data access (http://journals.plos.org/plosone/s/data-availability#loc-acceptable-data-sharing-methods).

Reviewers' comments:

Reviewer's Responses to Questions

**Comments to the Author**

1. Is the manuscript technically sound, and do the data support the conclusions?

Reviewer #1: Yes

Reviewer #2: Partly

Reviewer #3: Yes

2. Has the statistical analysis been performed appropriately and rigorously? 

Reviewer #1: Yes

Reviewer #2: No

Reviewer #3: Yes

3. Have the authors made all data underlying the findings in their manuscript fully available?

Reviewer #1: Yes

Reviewer #2: No

Reviewer #3: Yes

4. Is the manuscript presented in an intelligible fashion and written in standard English?

Reviewer #1: Yes

Reviewer #2: No

Reviewer #3: Yes

5. Review Comments to the Author

Reviewer #1: Re: The manuscript entitled “Lung cancer associated autoantibody responses are detected years before clinical presentation by Allen and co-workers.

Certainly, the study is interesting but as EarlyCDT lung test is commercial making the paragraphs nos. 5 and 6 below important. As the authors state it is previously described that autoantibodies appear long before clinical symptoms. It makes it necessary to define the new findings of the paper.

1. The objectives of the paper are unclear.

2. The authors state that previous work has shown that autoantibodies precede clinical presentation of lung cancer. Obviously, the current study confirms earlier findings. The Abstract must define what is new in the article.

3. What is the exact meaning of “clinical presentation” of lung cancer? The start of symptoms? The presentation by the family physician? The day of X-ray suspicion of malignant disease, or the day of the PAD diagnostic. Due to doctor’s delay, definitions of “presentation” may differ substantially thereby yielding different results.

4. Row 29. The sentence ending “with 7.2 years before it can be confirmed by imaging” needs a reference.

5. The EarlyCDT lung test is commercial. Obviously, the test is positive long before the appearance of radiology findings. If the surgeon does not have a visible tumour, it is impossible to operate. Thus, it is necessary to explain the clinical consequences of a positive finding. Frequent bronchoscopies?

6. The laboratory kit is for sale and the current findings make it tenable to consider its use for screening purposes. The authors must highlight and discuss its potentials as a screening tool. Which groups are suitable to screen? The number of smokers necessary to screen before finding a new lung cancer case? How to handle false positive samples?

7. The last sentence of the Discussion must be rephrased.

8. Obviously, the samples were run in 2016. For what reason is the publication delayed for almost 8 years?

Reviewer #2: I recommend rejecting this article for the following reasons:

- The sample size is low and lacks diversity, including only female participants from a single age category. This limitation impacts the generalization of the study's findings.

- Several studies showed that autoantibodies could be detected up to 5 years before lung cancer is diagnosed. The presented data is not considered as a novelty.

- The statistical methods used for sensitivity and specificity calculation are not correct.

- Tables 5, 6, and 7 are presented without adequate explanation.

- The authors did not mention the inclusion and exclusion criteria for the participants. In addition, there is no information on whether the patients were screened for the presence of autoimmune diseases.

- The discussion is poorly written. Adding other recent articles to support and compare the authors' findings would improve the discussion.

- The authors did not validate their funding in another independent dataset.

Reviewer #3: Overall a very good study indeed. Registry studies are important to generate evidences and for a long time they are being used in the HDI countries, it would be great if we consider this type of studies in LMICs too to make it more global and inclusive.

6. PLOS authors have the option to publish the peer review history of their article (what does this mean? ). If published, this will include your full peer review and any attached files.

**Do you want your identity to be public for this peer review?** For information about this choice, including consent withdrawal, please see our Privacy Policy .

Reviewer #1: No

Reviewer #2: **Yes: ** Sarra Mestiri

Reviewer #3: **Yes: ** Suvro Sankha Datta

---

## [Author Response · Author response to Decision Letter 1]

13 Sep 2024

Reviewer #1: Re: The manuscript entitled “Lung cancer associated autoantibody responses are detected years before clinical presentation by Allen and co-workers.

Certainly, the study is interesting but as EarlyCDT lung test is commercial making the paragraphs nos. 5 and 6 below important. As the authors state it is previously described that autoantibodies appear long before clinical symptoms. It makes it necessary to define the new findings of the paper.

1. The objectives of the paper are unclear.

Response: We appreciate the reviewers feedback, and have added to both the abstract and introduction to try and clarify the study objectives.

2. The authors state that previous work has shown that autoantibodies precede clinical presentation of lung cancer. Obviously, the current study confirms earlier findings. The Abstract must define what is new in the article.

Response: While there has been prior research showing elevated autoantibodies preceding cancer diagnoses, this is the first study to show that these autoantibody levels are detectable on a validated commercially available panel using commercially validated cutoff thresholds (against a clinical population of over 1600 subjects), we have now highlighted this novelty in the manuscript. Additionally, the previous studies were of lower sample size, being 49 cases and 54 controls (from an asbestosis population) in the case of Li et al., and 33 cases matched to 45 controls in the Lu et al. study, compared to the cohort of 142 cases and 142 controls explored in this study.

3. What is the exact meaning of “clinical presentation” of lung cancer? The start of symptoms? The presentation by the family physician? The day of X-ray suspicion of malignant disease, or the day of the PAD diagnostic. Due to doctor’s delay, definitions of “presentation” may differ substantially thereby yielding different results.

Response: Clinical presentation in this case referred to the date at which cancer diagnosis has been recorded in national cancer registries after histologic, cytologic, or clinical investigation had confirmed the presence of cancer. Manuscript wording has been updated to clarify this.

4. Row 29. The sentence ending “with 7.2 years before it can be confirmed by imaging” needs a reference.

Response: The sentence ending “with 7.2 years before it can be confirmed by imaging” is based on mathematical extrapolation of the previously referenced minimum tumour size for imaging detection of 100,000 cells (Fischer et al., 2006) which requires a tumour to double 16.6 times, and the mean doubling time for a malignant lung cancer of 158 days (Kanashiki et al., 2012), this then gives a time of 2622.8 days (16.6 multiplied by 158), or 7.2 years, for a tumour to grow to 100,000 cells.

5. The EarlyCDT lung test is commercial. Obviously, the test is positive long before the appearance of radiology findings. If the surgeon does not have a visible tumour, it is impossible to operate. Thus, it is necessary to explain the clinical consequences of a positive finding. Frequent bronchoscopies?

Response: There is indeed a potential issue with an autoantibody testing indicating a malignancy prior to imaging detection. Health economic analysis of results from the prospective phase IV ECLS trial has however shown that enhanced CT screening 6 monthly for 2 years in the case of a positive EarlyCDT test is cost effective, and would allow for the detection of tumours at the earliest point that they become visible. This detail has now been added to the discussion section of the manuscript, along with reference to the Health Economics analysis that has been published since the submission of this manuscript.

6. The laboratory kit is for sale and the current findings make it tenable to consider its use for screening purposes. The authors must highlight and discuss its potentials as a screening tool. Which groups are suitable to screen? The number of smokers necessary to screen before finding a new lung cancer case? How to handle false positive samples?

Response: The utility of the EarlyCDT test for screening is examined and discussed in greater depth in the reported results of the ECLS trial, a phase IV prospective trial which recruited high-risk patients, the most recent follow up data on this study has since been released as a preprint on medRxiv (https://www.medrxiv.org/content/10.1101/2024.06.13.24308919v1.full). Additionally details of the intended use populations for both screening and IPN risk stratification are detailed in the EarlyCDT Lung IFU. Greater detail of the ECLS trial and the use of the EarlyCDT test as a screening tool have been added to the discussion, although the principal aim of the reported study was to establish whether autoantibody responses preceded clinical presentation of lung cancer, and if so by how long.

7. The last sentence of the Discussion must be rephrased.

Response: While we are unsure as to the rationale behind the request to rephrase the last sentence, we have rephrased.

8. Obviously, the samples were run in 2016. For what reason is the publication delayed for almost 8 years?

Response: Freenome Ltd. (Formerly Oncimmune Ltd.) is a relatively small group, and unfortunately commercial priorities frequently impose upon our research goals, in the time since the data was initially made available to us we have undergone various commercial leadership changes, endured a global pandemic, gone through a period of restructure, and been purchased. None of these, however, impact upon the importance of the findings described in this study.

Reviewer #2: I recommend rejecting this article for the following reasons:

- The sample size is low and lacks diversity, including only female participants from a single age category. This limitation impacts the generalization of the study's findings.

Response: These limitations are acknowledged and discussed in the paper, and are a consequence of the difficulty and expense associated with sourcing adequate numbers of longitudinal samples in diseases with low prevalence. The sample size of 142 cases and 142 controls is much larger than the previously published work by Li et al. (49 cases and 54 controls – predominantly male from an asbestosis population) and Lu et al. (33 female breast cancer cases matched to 45 female controls), while the lack of male participants is unfortunate, establishment of diagnostic cutoff thresholds for the EarlyCDT test was completed on age and sex matched cohorts comprised of males and females, therefore we have no reason to expect the results reported here would differ in male subjects. The age range (50-75 year olds) represents a population with a higher risk of developing lung cancer, and is reflective of the intended use population of the EarlyCDT test (https://earlycdt.com/assets/ifus/IFU_EN_ECDTL2_Instructions%20for%20Use%20v14.pdf). The authors felt that in spite of the limitations inherent in the studied dataset, these findings were more robust than previously reported data, and the findings are of importance in demonstrating that elevated autoantibodies are present and measurable years before clinical presentation of lung cancer, and this validates the potential of autoantibody screening tests to identify cancer at the earliest stages when prognosis is vastly improved.

- Several studies showed that autoantibodies could be detected up to 5 years before lung cancer is diagnosed. The presented data is not considered as a novelty.

Response: We have now clarified the novelty of these findings in the manuscript, in that that this is the first time prediagnostic autoantibodies have been reported using a clinically validated autoantibody panel, using samples obtained from a prospective clinical trial. The reported results also used a larger cohort than previous studies, and sample positivity was determined using commercially used cutoffs defined during clinical validation studies, and not based on comparison of cases and controls within this study. This adds to the robustness of the reported results as no cutoff training was performed within the analysis.

- The statistical methods used for sensitivity and specificity calculation are not correct.

Response: The authors are unsure as to what statistical methods the reviewer is referring, sensitivity and specificity calculations are based upon standard contingency table calculations (detailed by Altman DG and Bland JM, Statistics Notes: Diagnostic tests 1: sensitivity and specificity. BMJ 1994;308:1552) and the numbers presented in Table 2, whereby:

Sensitivity = TP / (TP+FN) = 37 / (37+105) = 37 / 142 = 26.1%

Specificity = TN / (TN+FP) = 126 / (126 + 16) = 126 / 142 = 88.7%

- Tables 5, 6, and 7 are presented without adequate explanation.

Response: Explanation for tables 5, 6, and 7 was previously included in the patient cohort section and discussion, the authors have added contextual explanation to the results section as a preface to the tables.

- The authors did not mention the inclusion and exclusion criteria for the participants. In addition, there is no information on whether the patients were screened for the presence of autoimmune diseases.

Response: The manuscript states that the source of the samples was the UKCTOCS Study, eligibility and exclusion criteria for which are detailed in the Jacobs et al. paper referenced (Eligibility criteria were 50–74 years of age - aligning with the EarlyCDT intended use population - and postmenopausal status. Exclusion criteria were self-reported previous bilateral oophorectomy or ovarian malignancy, increased risk of familial ovarian cancer, or active non-ovarian malignancy). Smoking status was not part of the inclusion criteria, however only 20% of the cohort reported as never smokers. No screening was undertaken for autoimmune diseases, and autoimmune disease is not currently an exclusion criteria for the EarlyCDT Lung test.

- The discussion is poorly written. Adding other recent articles to support and compare the authors' findings would improve the discussion.

Response: The manuscript underwent review both externally and internally during it’s finalization, and no concerns were previously raised about the quality of the writing. Studies into prediagnostic cancer biomarkers, especially autoimmune based biomarkers, require large cohorts due to the low disease incidence, with follow up of patients over several years, this makes these studies logistically difficult, extremely expensive, and as a result extremely uncommon, and recent articles that could be used to support or contrast our findings do not exist.

- The authors did not validate their funding in another independent dataset.

Response: Again, this speaks to the sparsity of similar studies due to the logistical difficulty and expense associated with collecting adequate sample numbers for such longitudinal analyses in low population prevalence disease. As we mentioned before, clinical validation of the EarlyCDT test was performed independently of this cohort using a total of 1153 lung cancer cases and 937 normal controls, confirmed in eight separate patient groups totaling 451 lung cancer patients and 8831 normal subjects as well as an audit cohort of 1613 subjects (https://earlycdt.com/assets/ifus/IFU_EN_ECDTL2_Instructions%20for%20Use%20v14.pdf, Lam et al. 2011, Chapman et al. 2012, Jett et al. 2014), lending to the robustness of this data. Should an independent longitudinal dataset become available we would be extremely interested in validating these results. 

Reviewer #3: Overall a very good study indeed. Registry studies are important to generate evidences and for a long time they are being used in the HDI countries, it would be great if we consider this type of studies in LMICs too to make it more global and inclusive.

Response: We agree that further prospective longitudinal blood collection and disease monitoring studies would be beneficial in LMICs, and would be happy to collaborate with groups undertaking such studies.

---

## [Decision Letter · Decision Letter 1]

22 Nov 2024

Lung Cancer Associated Autoantibody Responses are Detectable Years Before Clinical Presentation

PONE-D-24-11623R1

Dear Dr. Allen,

We’re pleased to inform you that your manuscript has been judged scientifically suitable for publication and will be formally accepted for publication once it meets all outstanding technical requirements.

Kind regards,

Milad Khorasani, PhD

Academic Editor

PLOS ONE

Additional Editor Comments (optional):

Reviewers' comments:

Reviewer's Responses to Questions

**Comments to the Author**

1. If the authors have adequately addressed your comments raised in a previous round of review and you feel that this manuscript is now acceptable for publication, you may indicate that here to bypass the “Comments to the Author” section, enter your conflict of interest statement in the “Confidential to Editor” section, and submit your "Accept" recommendation.

Reviewer #1: All comments have been addressed

Reviewer #3: All comments have been addressed

2. Is the manuscript technically sound, and do the data support the conclusions?

Reviewer #1: Yes

Reviewer #3: Yes

3. Has the statistical analysis been performed appropriately and rigorously? 

Reviewer #1: Yes

Reviewer #3: Yes

4. Have the authors made all data underlying the findings in their manuscript fully available?

Reviewer #1: Yes

Reviewer #3: Yes

5. Is the manuscript presented in an intelligible fashion and written in standard English?

Reviewer #1: Yes

Reviewer #3: Yes

6. Review Comments to the Author

Reviewer #1: (No Response)

Reviewer #3: Appreciate your effort in responding to the comments made by the reviewers. Overall, I found that it is OK.

7. PLOS authors have the option to publish the peer review history of their article (what does this mean? ). If published, this will include your full peer review and any attached files.

**Do you want your identity to be public for this peer review?** For information about this choice, including consent withdrawal, please see our Privacy Policy .

Reviewer #1: **Yes: ** Petter Järemo MD, PhD

Reviewer #3: No

---

## [Editor Report · Acceptance letter]

PONE-D-24-11623R1

PLOS ONE

Dear Dr. Allen,

I'm pleased to inform you that your manuscript has been deemed suitable for publication in PLOS ONE. Congratulations! Your manuscript is now being handed over to our production team.

Kind regards,

on behalf of

Dr. Milad Khorasani

Academic Editor

PLOS ONE